# The Effect of COVID-19 Pandemic on Overall and Cause-Specific Mortality in Pavia, Northern Italy: Updated Estimates for the Year 2021

**DOI:** 10.3390/ijerph20085529

**Published:** 2023-04-17

**Authors:** Paola Bertuccio, Pietro Perotti, Giansanto Mosconi, Simona Dalle Carbonare, Federica Manzoni, Lorenza Boschetti, Stefano Marguati, Paolo Paraluppi, Lorenzo Blandi, Leandro Gentile, Maddalena Gaeta, Lorella Cecconami, Anna Odone

**Affiliations:** 1Department of Public Health, Experimental and Forensic Medicine, University of Pavia, 27100 Pavia, Italy; 2Health Protection Agency of Pavia (ATS Pavia), 27100 Pavia, Italy

**Keywords:** COVID-19, pandemic burden, population impact, excess mortality, cause of death, regression models

## Abstract

Excess mortality estimates are considered relevant indicators of direct and indirect pandemic effects on the population. Scant data have been published on cause-specific excess mortality. Using individual-level administrative data covering the Pavia province of Italian northern Lombardy region, we provided all-cause and cause-specific raw (RMR) and age-standardized (ASMR) mortality rates in 2021 and 2015–2019, the rate ratio, and 95% confidence intervals, overall and by sex. We obtained the excess deaths in 2021 as the difference between the number of observed and expected deaths from all causes and the two leading causes of death (all neoplasms and circulatory system diseases) by fitting over-dispersed quasi-Poisson regression models, accounting for temporal, seasonal and demographic changes. The total ASMR in 2021 was 972.4/100,000 (6836 certified deaths), with the highest ASMRs for circulatory system diseases (272.6/100,000) and all neoplasms (270.3/100,000), followed by COVID-19 (94.8/100,000 and 662 deaths). Compared to the expected, we estimated a total of 6.2% excess deaths in 2021 (7.2% in males and 5.4% in females), with no excess deaths from all neoplasms and a 6.2% reduction from circulatory system diseases. COVID-19 continued to affect total mortality in 2021, albeit to a lesser extent than in 2020, consistently with national patterns.

## 1. Introduction

Coronavirus disease 2019 (COVID-19), an infectious disease caused by the severe acute respiratory syndrome coronavirus 2 (SARS-CoV-2), was declared a pandemic on 11 March 2020 by the World Health Organization (WHO) [1,2]. Quantifying the burden and population impact of the COVID-19 pandemic is challenging [3]. Direct measures of COVID-19 morbidity and mortality are affected by varying testing rates, capabilities and strategies, heterogeneous COVID-19 death definitions and surveillance systems [4,5,6]. Overall excess mortality is considered a key measure to understand, compare and monitor the impact of the COVID-19 pandemic over time, both from an epidemiological and public health perspective [7,8]. To this end, death certificates represent the most reliable source of information to compare cause-specific mortality across countries.

The WHO defined excess mortality as “the mortality above what would be expected based on the non-crisis mortality rate in the population of interest” [9]. Previous national [10,11,12,13] and international [14,15,16,17] studies estimated excess mortality due to COVID-19, taking advantage of different statistical methods. Some earlier studies made simple comparisons between mortality rates registered in a period before the pandemic (usually an average of the previous five years) [10] and those registered after the pandemic. Other studies [11,12,18] performed complex regression models to take into account factors that could affect mortality rates, including seasonality, air temperature, sex and age. Italy is among the European countries that were hit first and hardest by the pandemic, with dramatic consequences in terms of morbidity and mortality, although heterogeneous across its territory. The Lombardy region, in the North of the country, was the epicentre of the outbreak and accounted for the largest share of total COVID-19 deaths [4,19]. Most of excess deaths occurred in 2020, when vaccines were not yet available, and more severe SARS-CoV-2 variants were circulating [20,21], with approximately 100,000 estimated excess deaths in Italy [12]; 35,000 of these were in Lombardy [22]. In 2021, thanks to mass vaccination uptake in the adult population, and less virulent variants of COVID-19, excess mortality decreased, with almost 35,000 estimated excess deaths at the national level and 4000 in Lombardy [12,22].

Interest on excess mortality analysis has grown in recent times, together with the application of more complex statistical approaches [9]. Whereas data on excess mortality from all causes are available, most studies did non explore excess mortality by specific causes. We previously published all-cause and cause-specific mortality estimates in Pavia province in Lombardy region for 2020, using individual-level administrative data from the local Health Protection Agency (HPA) [23]. 2020 excess total mortality in Pavia has been 24% in men and 25% in women. Consistent with other published studies [10,24], significant increases were found for infectious and parasitic diseases, respiratory system diseases, dementia and Alzheimer’s disease. Conversely, reductions in mortality were observed for all neoplasms, circulatory system diseases, and transport accidents. The current study aims to update all-cause and cause-specific mortality rates to the year 2021 in Pavia province and to apply advanced statistical modelling to explore overall and cause-specific excess deaths patterns.

## 2. Materials and Methods

We used individual-level administrative data on certified deaths and resident population in the province of Pavia over the decade 2011–2021. In detail, copies of death certificates of residents in Pavia province are transmitted to the HPA for coding the underlying cause of death according to the Tenth Revision of the International Classification of Diseases (ICD-10). In accordance with the latest guidelines by the WHO [25], deaths attributed to COVID-19 were defined as those in which COVID-19 was the underlying cause of death. Pavia HPA staff performed data entry in the mortality database: an ad-hoc developed software used in Lombardy region automatically codes 90% of records, and the remaining records (mostly from external cases, childhood mortality and other peculiar causes) are manually coded.

### Statistical Analysis

For all-cause mortality and selected groups of causes, we computed raw mortality rates (RMR) and age-standardized mortality rates (ASMR) using the direct method (2011 Italian population as standard) for the quinquennium 2015–2019 and in 2021, in the total population, and in males and females, separately. We computed the rate ratio (RR) and corresponding 95% confidence intervals (CI) between 2021 and 2015–2019 ASMRs. 

Then, focusing on deaths from all causes and the two major causes of death in Pavia province, i.e., all neoplasms (ICD 10 codes: C00-D48) and circulatory system diseases (ICD 10 codes: I00-I99), we estimated the number of expected deaths in 2021, by fitting separate quasi-Poisson regression models based on mortality and population data over the period 2011–2019 plus January–February 2020, in males and females, separately. The model included the number of deaths as the response variable and a linear term for each calendar year (to take into account temporal trends), five-years age groups (to capture demographic changes over time), and calendar month (to capture seasonal variability), plus the natural logarithm of the population as offset term. Thus, we obtained the number of excess deaths in 2021 as the difference between observed and expected deaths. We also computed the 95% CI for the excess number of deaths through a Monte Carlo simulation based on an iterative approach. We randomly ran 10,000 samples (i.e., number of iterations) from a multivariate normal distribution for each set of the model’s coefficients using the parameter estimates and the variance-covariance matrix. For each sample, from the distribution of the difference between observed and expected deaths, we obtained the 95% CIs using normal approximation.

## 3. Results

Table 1 reports RMR and ASMR from all causes and selected groups of causes, along with the RR between ASMR in 2021 and the 2015–2019 and corresponding 95% CI. Figure 1 illustrates the number of certified deaths registered in 2021 in Pavia province, ordered from the highest to the lowest number of deaths in males and females, separately. In 2021, ASMR from all causes in Pavia was 972.4/100,000 (958.1 in males and 977.5 in females), corresponding to 6836 registered deaths (3174 males and 3662 females). The leading causes of death were circulatory system diseases, with over 2 thousand deaths (ASMR 272.6/100,000), and all neoplasms, with 1760 deaths (ASMR 270.3/100,000). All neoplasms were the primary cause of deaths in males (955 deaths, ASMR 299.9/100,000), followed by circulatory system diseases (857 deaths, ASMR 250.4/100,000). In females, the primary cause of death were circulatory system diseases (1147 deaths, ASMR 291.8/100,000), followed by all neoplasms (808 deaths, ASMR 239.4/100,000). COVID-19 was the third leading cause of death in both sexes, with 337 deaths among males (ASMR 101.2/100,000) and 325 among females (ASMR 87.5/100,000). Overall, 353 deaths due to dementia and Alzheimer’s disease were registered in 2021 (ASMR 46.3/100,000), more among females than males (252 vs. 101). 

Looking at the comparison between 2021 and 2015–2019 ASMRs (Table 1), significant decreases in mortality emerged for all neoplasms (−10%), dementia and Alzheimer’s disease (−15%), circulatory system diseases (−17%), influenza and pneumonia (−26%), and chronic lower respiratory diseases (−27%). These reductions remained significant among males, excluding dementia and Alzheimer’s disease, while they were observed only for circulatory system diseases among females.

Table 2 and Table 3 report total number of—respectively—all-cause and cause-specific (all neoplasms and circulatory system diseases) observed, expected and excess deaths in 2021, by sex and age. In 2021, we estimated an excess of 400 (95% CI: 253–546) total deaths—thus +6.2% (95% CI: 3.9–8.5) over the 6436 expected deaths. Observed deaths were higher than expected in all age groups, with estimated excesses of 7.8% (95% CI: 5.0–10.5) at age 65–79 years, and 5.9% (95% CI: 3.6–8.3) at age 80 or more years. The excess deaths were 7.2% (95% CI: 3.8–10.6) among males and 5.4% (95% CI: 2.2–8.5) among females (Table 2).

With reference to excess deaths by cause (Table 3), no significant excesses emerged in mortality from all neoplasms. Of note, a significant excess mortality of 11.7% (95% CI: 3.7–19.6) was registered among females aged 80 years and over.

A significant lower number of deaths in 2021 was estimated from circulatory system diseases compared to the expected ones (−6.1%; 95% CI: −9.9–−2.3), more evident among females (−9.0%; 95% CI: −14.0–−4.0), than in males (−1.9%; 95% CI: −7.9–4.0). In the population aged 25–64, a marked excess in deaths was estimated among both males (15.1%; 95% CI: 5.5–23.3) and females (26.3%; CI: 10.5–36.8). Conversely, deaths were 10.9% (95% CI: −15.9–−5.8) lower than expected among women aged 80 and over.

## 4. Discussion

This study updated estimates of excess in all-cause and cause-specific mortality in 2021 in Pavia province, based on official certified data provided by the local HPA. The impact of the COVID-19 pandemic on excess mortality in Italy has been investigated in several studies through various methodological approaches [10,12,13,26]. The originality of this work is that, focusing on all causes and the two leading causes of death (i.e., all neoplasms and circulatory system diseases) in the area, we were able to provide more accurate excesses’ estimates by using an advanced statistical approach. This allowed us to consider temporal, seasonal and demographic changes over a long pre-pandemic period, i.e. from 2011 to 2019. Compared to the expected, we estimated about 7% excess deaths in males and about 5% in females. In males, neoplasms were the first cause of death, followed by circulatory system diseases, while the opposite was reported in females. For both sexes, the third most frequent cause of death remained COVID-19. The observed number of deaths from all neoplasms was approximately in line with the expected, whereas deaths from circulatory system disorders were −6% lower than expected. 

According to the WHO, global excess mortality has more than doubled in 2021 compared to 2020 [9]. The reasons for this increase have been attributed to the fact that, in 2021 the SARS-CoV-2 infection has largely spread to densely populated countries where there had been only limited exposure in 2020 and access to COVID-19 vaccines was scarce or delayed. The pandemic burden in such locations has also been worsened by the emergence of more infectious and lethal SARS-CoV-2 variants, like the Delta variant [9]. Changes in variants’ spread likely determines a huge variability in excess mortality patterns [9], as emerged by a paper based on Chinese data [27]. 

Italy was one of the first countries outside China where large outbreaks of COVID-19 occurred in the first phase of the pandemic [4], recording an excess mortality of over 15% in 2020 compared to the pre-pandemic period 2015–2019 [14]. However, in 2021, nationwide, the total number of deaths from all causes, although still higher (i.e., almost 10%) than the 2015–2019 average, was 5% lower than the previous year [12,28]. Italy’s resilience to the spread of the Delta variant of SARS-CoV-2 is likely due to early rollout of COVID-19 vaccines in the country [29]. The Italian COVID-19 mass immunization campaign began on 27 December 2020, and initially targeted healthcare personnel and frail and older individuals [30]. Indeed, a large portion of excess mortality in 2021 compared to 2015–2019 occurred in the first four months when adult population’s vaccine coverage was still low [28]. Conversely, from the summer of 2021 onwards, the high vaccination coverage level has been matched by a sharp decline in COVID-19-related fatalities [28].

Compared to the 2015–2019 average, for the year 2021, an excess of mortality was recorded all over the country, albeit with a South-North gradient [28]. While in the South of Italy—least affected by the pandemic in 2020—the 2021 excess mortality was over 13%, in the North—hit first by the pandemic in 2020 [4,22,31]—2021 excess mortality was just over 8% [28]. According to our data, in the province of Pavia, excess mortality in 2021 was significantly lower than the previous year [23], settling at levels quite below than those recorded on average in the North of Italy.

In line with other studies [32,33], we found that, at working age, 2021 excess mortality was greater in males than in females. In accordance with national data [28], the main contribution to 2021 excess mortality came from the increase in deaths of people aged over 80 years.

In Italy, as in most high-income countries, the ratio of the number of deaths from circulatory system diseases to the number of deaths from all neoplasms among adults increases with age [34]. Since most of deaths from COVID-19 occurred in people over the age of 80, it is likely that the harvesting effect had a greater impact on mortality from circulatory system diseases than on mortality from cancer [35]. In fact, our age-stratified estimates showed significant reductions in deaths—as compared to the expected ones—in the 65–79 age group for neoplasms, while in the oldest age group (i.e., 80 or more) for cardiovascular causes. However, overall excess deaths from all neoplasms in all age groups was in line with expectations. This finding is confirmed by other studies which reported cancer patients experienced lower or similar mortality rates compared to those observed among healthy people during the pandemic [36,37]. 

Besides the mortality displacement, other reasons can likely contribute to attenuate the COVID-19 impact on mortality from neoplasms, and circulatory system diseases in selected age groups. Although these conditions have been linked to higher lethality in COVID-19 patients admitted to hospital [38], there is evidence that individuals with chronic diseases, perhaps aware of their higher risks or simply because less socially active, may have somewhat protected themselves from opportunities for contagion [39]. Nevertheless, long-term pandemic effects on cancer, but also cardiovascular diseases, needs to be further explored in the future.

For cardiovascular causes, the increase found at age group 25–64 could be explained by the so-called long-COVID-19 [40,41]. According to previous research, younger individuals, who are generally more protected from adverse outcomes in the acute phase of COVID-19, still show an increased risk of cerebrovascular disorders, dysrhythmias, ischemic and non-ischemic heart disease, pericarditis, myocarditis, heart failure, thromboembolic disease as early as the first year after infection [41].

Studies on excess mortality and the estimate of its excess due to the pandemic have accumulated over these last two years, applying different methodological approaches [17,18,22]. Our findings should be interpreted with caution. We should consider that the COVID-19 disease could led to the final event of death especially in people with comorbidities as compared to people without other conditions than COVID-19 [38]. This could cause a possible misclassification, both under- and over-reporting of death from COVID-19, and other causes as well. However, this analysis is based on death certificates, which represent the primary source of epidemiologic data, on the entire population, since the coding system of death (through the ICD) is highly standardized and internationally adopted. 

Our study, although based on local data, has some advantages. First, we used complete and official certified data covering the whole province of Pavia, located in the Lombardy region, that was the epicenter of the pandemic in Italy. Second, estimates of expected deaths are based on statistical models that allowed us to take into account changes in age structure of the population, seasonal variability and temporal improvements of mortality trends, over a long period before the pandemic. To our knowledge, this is the first study, based on data covering a whole Italian northern province, which estimated excess mortality not only from all causes combined but also from the major causes in 2021, through a statistical model. Our report can be informative for health policies, not only at local level but also for regional and national interest. 

## 5. Conclusions

The COVID-19 pandemic differentially affected health outcomes, and analysis of mortality data detailed for specific causes or groups of causes can help to better investigate both direct and indirect COVID-19 pandemic effects. To better monitor pandemic effects and possible future health emergencies, it is essential to strengthen timely death registration systems, long recognized as critical to a successful public health strategy. 

Further investigations are needed to help distinguish between changes in excess mortality directly associated to SARS-CoV-2 infection, and those indirectly caused by the pandemic. This would also help to understand the reasons for the heterogeneity in the lethality of COVID-19 in different communities, as well as hopefully obtain useful elements to predict trends of possible future pandemics and more consciously screen different health policy options.

## Figures and Tables

**Figure 1 ijerph-20-05529-f001:**
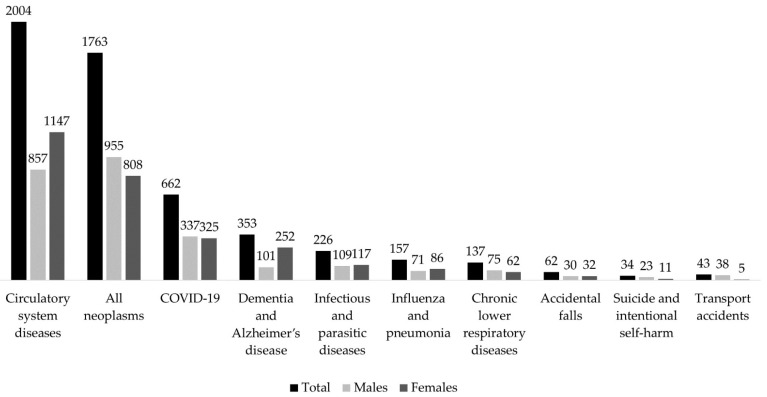
Bar plots with number of certified deaths from the first ten causes, registered in 2021, ordered from the highest to the lowest numbers, in the total, males and females population. Pavia province, Lombardy region, Italy.

**Table 1 ijerph-20-05529-t001:** Number of observed deaths, raw mortality rates (RMR) and age-standardized ^1^ mortality rates (ASMR/100,000) from all causes and selected groups of causes, in the total, males and females population, at all ages, in 2015–2019 and 2021, the Rate Ratio between ASMRs in 2021 and 2015–2019 and corresponding 95% confidence interval (CI). Pavia province, Lombardy region, Italy.

	2015–2019			2021			Rate Ratio (95% CI)
Group of Causes	Average Deaths	RMR	ASMR	Deaths	RMR	ASMR	
**Total**							
All causes	6653	1215.9	978.3	6836	1275.9	972.4	0.99 (0.96–1.03)
COVID-19	-	-	-	662	123.6	94.8	-
Infectious and parasitic diseases (COVID-19 excluded)	186	34.0	27.8	226	42.2	32.1	1.15 (0.95–1.40)
**All neoplasms**	1926	352.1	300.9	1763	329.0	270.3	**0.90 (0.84–0.96)**
Diabetes mellitus	173	31.7	25.5	157	29.3	21.5	0.85 (0.68–1.05)
**Dementia and Alzheimer’s disease**	393	71.9	54.2	353	65.9	46.3	**0.85 (0.74–0.99)**
**Circulatory system diseases**	2326	425.2	329.3	2004	374	272.6	**0.83 (0.78–0.88)**
**Influenza and Pneumonia**	207	37.9	29.0	157	29.3	21.4	**0.74 (0.60–0.91)**
**Chronic lower respiratory diseases**	181	25.6	18.9	137	33.2	26	**0.73 (0.58** **–0.91)**
Transport accidents	32	5.9	5.7	43	8.0	7.7	1.33 (0.84–2.11)
Accidental falls	66	12.0	9.4	62	11.6	8.3	0.88 (0.62–1.25)
Suicides and intentional self-harm	43	6.4	5.9	34	6.4	5.9	0.78 (0.50–1.23)
**Males**							
All causes	3054	1146.0	974.7	3174	1210.2	958.1	0.99 (0.93–1.03)
COVID-19	-	-	-	337	128.5	101.2	-
Infectious and parasitic diseases (COVID-19 excluded)	88	33.0	28.1	109	41.6	32.5	1.16 (0.87–1.54)
**All neoplasms**	1060	397.9	345.8	955	364.1	299.9	**0.87 (0.80–0.95)**
Diabetes mellitus	80	30.0	25.5	66	25.2	19.5	0.76 (0.55–1.06)
Dementia and Alzheimer’s disease	112	42.0	34.1	101	38.5	28.9	0.85 (0.65–1.11)
**Circulatory system diseases**	953	357.6	298.2	857	326.8	250.4	**0.84 (0.77–0.92)**
**Influenza and Pneumonia**	93	35.1	28.9	71	27.1	20.3	**0.70 (0.51–0.96)**
**Chronic lower respiratory diseases**	97	36.5	29.8	75	28.6	21.4	**0.72 (0.53–0.97)**
Transport accidents	26	9.6	9.4	38	14.5	13.6	1.45 (0.87–2.41)
Accidental falls	29	11.0	9.2	30	11.4	8.7	0.95 (0.57–1.59)
Suicide and intentional self-harm	33	12.5	11.8	23	8.8	8.1	0.68 (0.40–1.17)
**Females**							
All causes	3599	1282.3	983.1	3662	1338.8	977.5	0.99 (0.95–1.04)
COVID-19	0	-	-	325	118.8	87.5	-
Infectious and parasitic diseases (COVID-19 excluded)	98	35	27.7	117	42.8	31.4	1.14 (0.87–1.49)
All neoplasms	866	308.6	259.1	808	295.4	239.4	0.92 (0.84–1.02)
Diabetes mellitus	93	33.3	25.4	91	33.3	23.3	0.92 (0.69–1.23)
Dementia and Alzheimer’s disease	282	100.3	72.7	252	92.1	62.3	0.86 (0.72–1.02)
**Circulatory system diseases**	1373	489.4	358.5	1147	419.3	291.8	**0.81 (0.75–0.88)**
Influenza and Pneumonia	114	40.6	29.4	86	31.4	22.5	0.76 (0.58–1.02)
Chronic lower respiratory diseases	84	30.0	22.8	62	22.7	16.4	0.72 (0.52–1.01)
Transport accidents	6	2.3	2.2	5	1.8	1.8	0.79 (0.24–2.58)
Accidental falls	37	13.2	9.8	32	11.7	7.8	0.80 (0.50–1.30)
Suicide and intentional self-harm	10	3.5	3.3	11	4.0	3.6	1.10 (0.46–2.62)

RMR: raw mortality rate; ASMR: age-standardized mortality rates. ^1^ Italian standard population, Census 2011. The rate ratios statistically significant are highlighted in bold.

**Table 2 ijerph-20-05529-t002:** Number of observed and expected deaths, along with the excess mortality estimates in 2021, for all causes, at all ages and three age groups, in the total, males and females population. Pavia province, Lombardy region, Italy.

	Observed Deaths	Expected ^1^ Deaths	Excess Deaths (95% C.I.)	% Excess Deaths (95% C.I.)
**All causes**				
**Total**				
All ages	6836	6436	400 (253–546)	6.2 (3.9–8.5)
25–64	639	614	25 (3–46)	4.1 (0.5–7.5)
65–79	1609	1493	116 (74–157)	7.8 (5.0–10.5)
80+	4565	4309	256 (154–357)	5.9 (3.6–8.3)
**Males**				
All ages	3174	2960	214 (112–315)	7.2 (3.8–10.6)
25–64	414	389	25 (7–42)	6.4 (1.8–10.8)
65–79	983	894	89 (54–123)	10.0 (6.0–13.8)
80+	1762	1664	98 (38–157)	5.9 (2.3–9.4)
**Females**				
All ages	3662	3476	186 (77–294)	5.4 (2.2–8.5)
25–64	225	224	1 (−11–13)	0.4 (−4.9–5.8)
65–79	626	598	28 (4–51)	4.7 (0.7–8.5)
80+	2803	2645	158 (74–241)	6.0 (2.8–9.1)

^1^ Estimated obtained by quasi-Poisson regression models on mortality and population data over the period 2011–2019 (plus January–February 2020), separately by sex. The models included the number of deaths as the response variable and a linear term for each calendar year (to take into account temporal trends in mortality), five-years age groups (to capture demographic changes over time), and calendar month (to capture seasonal variability), plus the natural logarithm of the population as offset term.

**Table 3 ijerph-20-05529-t003:** Number of observed and expected deaths, along with the excess mortality estimates in 2021, for all neoplasms and circulatory system diseases, at all ages and three age groups, in the total, males and females population. Pavia province, Lombardy region, Italy.

	Observed Deaths	Expected ^1^ Deaths	Excess Deaths (95% C.I.)	% Excess Deaths (95% C.I.)
**All neoplasms**				
**Total**				
All ages	1763	1761	2 (−109–113)	0.1 (−6.2–6.4)
25–64	298	279	19 (−1–39)	6.8 (−0.4–14.0)
65–79	631	678	−47 (−92–−1)	−6.9 (−13.6–−0.1)
80+	830	794	36 (−15–87)	4.5 (−1.9–11.0)
**Males**				
All ages	955	983	−28 (−106–50)	−2.8 (−10.8–5.1)
25–64	165	156	9 (−6–24)	5.8 (−3.8–15.4)
65–79	385	412	−27 (−62–8)	−6.6 (−15.0–1.9)
80+	402	410	−8 (−42–26)	−2.0 (−10.2–6.3)
**Females**				
All ages	808	778	30 (−29–89)	3.9 (−3.7–11.4)
25–64	133	123	10 (−1–21)	8.1 (−0.8–17.1)
65–79	246	266	−20 (−41–1)	−7.5 (−15.4–0.4)
80+	428	383	45 (14–75)	11.7 (3.7–19.6)
**Circulatory system diseases**				
**Total**				
All ages	2004	2135	−131 (−211–−50)	−6.1 (−9.9–−2.3)
25–64	108	92	16 (8–23)	17.4 (8.7–25.0)
65–79	360	350	10 (−6–26)	2.9 (−1.7–7.4)
80+	1536	1688	−152 (−216–−87)	−9.0 (−12.8–−5.2)
**Males**				
All ages	857	874	−17 (−69–35)	−1.9 (−7.9–4.0)
25–64	84	73	11 (4–17)	15.1 (5.5–23.3)
65–79	220	214	6 (−8–20)	2.8 (−3.7–9.3)
80+	553	584	−31 (−67–5)	−5.3 (−11.5–0.9)
**Females**				
All ages	1147	1261	−114 (−176–−51)	−9.0 (−14.0–−4.0)
25–64	24	19	5 (2–7)	26.3 (10.5–36.8)
65–79	140	136	4 (−5–13)	2.9 (−3.7–9.6)
80+	983	1103	−120 (−175–−64)	−10.9 (−15.9–−5.8)

^1^ Estimated obtained by quasi-Poisson regression models on mortality and population data over the period 2011–2019 (plus January–February 2020), separately by sex. The models included the number of deaths as response variable and a linear term for each calendar year (to take into account for temporal trends in mortality), five-years age groups (to capture demographic changes over time), and calendar month (to capture seasonal variability), plus the natural logarithm of the population as offset term.

## Data Availability

Not applicable.

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
