# Peer review of "The Effect of COVID-19 Pandemic on Overall and Cause-Specific Mortality in Pavia, Northern Italy: Updated Estimates for the Year 2021"

_ijerph, 2023, doi:10.3390/ijerph20085529_

Round 1
Reviewer 1 Report
My sincere congratulations to the authors for the paper. Nothing to say from a methodological point of view. All the sections are correct. I just have a curiosity rather than a doubt, are you sure you have not encountered any limitations in this study?
Author Response
Thank you for the positive feedback: we are glad the reviewer appreciate the paper and specifically its methodological aspects. We have added a paragraph discussing some limitations in the Discussion.

Reviewer 2 Report
The paper is well-written, and I am happy with the authors' statistical approach.
The author should have a sub-heading for “Statistical analysis” in the manuscript.
My only question is that, in Table 1, I am not sure why the authors have the estimate for 2015-2019 and their objective was “to update all-cause and cause-specific mortality rates estimates to the year 2021 in Pavia province” and on the “Impact of the COVID-19 pandemic.”
Did we not have Covid-19 in 2015-2018? The impact did not start until early 2020 – I think the authors need to justify this in the paper's discussion section. Or on the alternative, I would suggest that the authors take the 2015-2019 estimate out of table 1 and remove the “Impact of the COVID-19 pandemic” from the title because they did not show any impact in figure 1 (comparing pre and post-Covid-19) rather they ONLY showed results for 2021.
Author Response
Thank you for reviewing our manuscipt and for the appreciation. Please find attached a point-by-point response to the comments.

Reviewer 3 Report
This is a 2021 follow up to the article regarding cause-specific mortality due to COVID-19 in 2020 produced by the same lead authors: Perotti, P.; Bertuccio, P.; Cacitti, S.; Deandrea, S.; Boschetti, L.; Dalle Carbonare, S.; Marguati, S.; Migliazza, S.; Porzio, E.; Riboli, S.; Cadum, E.; Cecconami, L.; Odone, A. Impact of the COVID-19 Pandemic on Total and Cause-Specific Mortality in Pavia, Northern Italy. Int. J. Environ. Res. Public Health 2022, 19, 6498. https://doi.org/10.3390/ijerph19116498. In this regard, the submission of 2021 follows the same outline as the paper on the 2020 results. Therefore, it would seem reasonable to publish this work as is. The problem is that there are too many unanswered questions in this submission that should be responded to before this paper is considered for publication.
The strengths of this paper are the clarity of: the argument presented, the method, and presentation of the results. The weaknesses are that there is no explanation of what is COVID-19 and no definition of COVID-19-related deaths. Furthermore, these represent unanswered questions: What was the effect of the variants in 2021 being less severe than those of 2020? What is the relationship between COVID-19 and circulatory problems as the cause of death? How do these results compare with those of China and worldwide?
As per the instructions available on the Word template for IJERPH, all citations should be in square brackets, not round brackets. Furthermore, the references must be reformatted to correspond to the template instructions. This includes that the DOI number should be added when available.
Line by line suggested edits.
12-33 Abstracts are expected to be no more than 200 words for IJERPH. This abstract is 327 words. Please reduce the number of words to 200.
34 It is suggested that the authors add “pandemic burden” as the second keyword and “population impact” as the third keyword.
37 Before starting on the topic of this paper, please provide a sentence with a reference on what is COVID-19 and that is was declared a pandemic on 11 March 2020 by the WHO.
45 Please indent the first sentence.
59 Change “35,000 in Lombardy” to “35,000 of these were in Lombardy”.
60 Given that the authors have said in line 58 that more severe variants of COVID-19 were circulating in 2020 than in 2021, change “population, excess” to “population, and less virulent variants of COVID-19, excess”.
68 Change “Consistently” to “Consistent”.
80-81 “the latest recommendations by the WHO”—please provide a reference to these latest recommendations.
95 Change “as response” to “as the response”.
96 Change “account for temporal” to “account temporal”.
101 Change “run” to “ran”.
120-121 When the authors say that COVID-19 was the third leading cause of death, can they be more specific regarding what was the presentation of the cause of death? For example, was the death caused by problems with respiration or by a circulatory problem brought on the by infection?
125-130 Is it possible that there really was no decrease in mortality in relation to each of these causes, rather, instead of listing these as the cause of death, COVID-19 was listed as the cause of death if the patient merely tested positive for the disease? Is it possible that It wasn’t COVID-19 that was the cause of death, it was, instead, one of these other reasons that was found to decrease? The authors are asked to discuss this possibility. Here are two articles to consider in this regard:
Polverino, F.; Stern, D.A.; Ruocco, G.; Balestro, E.; Bassetti, M.; Candelli, M.; Cirillo, B.; Contoli, M.; Corsico, A.; D'Amico, F.; D'Elia, E. Comorbidities, cardiovascular therapies, and COVID-19 mortality: a nationwide, Italian Observational Study (ItaliCO). Frontiers Cardiovascular Med. 2020, 7, p.585866. https://doi.org/10.3389/fcvm.2020.585866
Onder, G.; Rezza, G.; Brusaferro, S. Case-fatality rate and characteristics of patients dying in relation to COVID-19 in Italy. Jama 2020, 323, 1775-1776. https://doi.org/10.1001/jama.2020.4683.
151-157 Table 1
It is very interesting that the deaths by suicides and intentional self-harm decreased during 2021 according to this research. It has been claimed by other researchers that COVID-19 increased suicides in Northern Italy during this time. Please read this reference and discuss why there is a discrepancy regarding reported suicide of the current study with the research presented in this article.
Calati, R.; Gentile, G.; Fornaro, M.; Madeddu, F.; Tambuzzi, S.; Zoja, R. (2023). Suicide and homicide before and during the COVID-19 pandemic in Milan, Italy. J. Affect. Disord. Rep. 2023, 100510. https://doi.org/10.1016/j.jadr.2023.100510.
160-162 Figure 1
The font used is too small to read. Please enlarge these three bar plots and place them consecutively, in vertical arrangement. The title of figures goes below the figure, not above according to the Word template.
169 Change “as response” to “as the response”.
196-198 The authors had stated previously in line 58 that the variants of COVID-19 in 2021 were less severe than in 2020. Was this true of the variants that circulated in 2021 in densely populated countries that had limited exposure in 2020? The authors should discuss why it might be that excessed mortality more than doubled in 2021 from that of 2020 even though vaccines had been developed and the variants that were circulating were less severe.
198 Given that Italy was one of the first countries outside China were large outbreaks of COVID-19 occurred in 2020, it would be relevant to compare the excess deaths in Italy in 2021 with those in China to see if the trend observed in Italy was similar to China. Here are references to assist in this.
Lin, J.; Huang, G.; Wei, Y.; Pei, L. Measuring the Effect of COVID-19 Pandemic on Mortality: Review and Prospect—China, 2021. China CDC Weekly 2022, 4, 499. https://doi.org/10.46234/ccdcw2022.110.
Ioannidis, J.P A.; Zonta, F.; Levitt, M. Estimates of COVID-19 deaths in Mainland China after abandoning zero COVID policy. Euro. J. Clin. Invest. 2023, e13956–e13956. https://doi.org/10.1111/eci.13956.
Furthermore, it would be good to situate these results with those that have been established worldwide. See:
Shang, W.; Wang, Y.; Yuan, J.; Guo, Z.; Liu, J.; Liu, M. Global Excess Mortality during COVID-19 Pandemic: A Systematic Review and Meta-Analysis. Vaccines 2022, 10, 1702. https://doi.org/10.3390/vaccines10101702.
Msemburi, W.; Karlinsky, A.; Knutson, V.; Aleshin-Guendel, S.; Chatterji, S.; Wakefield, J. The WHO estimates of excess mortality associated with the COVID-19 pandemic. Nature 2023, 613, 130–137. https://doi.org/10.1038/s41586-022-05522-2.
He, G., Xiao, J., Lin, Z., & Ma, W. Excess mortality, rather than case fatality rate, is a superior indicator to assess the impact of COVID-19 pandemic. Innovation 2022, 3, 100298. https://doi.org/10.1016/j.xinn.2022.100298
240 Change “at age” to “for age”.
254 “the first study in Italy which estimated cause-specific excess mortality in 2021”, please indicate in the discussion how this current study differs from this other published work.
Alicandro, G.; Remuzzi, G.; Centanni, S.; Gerli, A.; La Vecchia, C. (2022). Excess total mortality during the Covid-19 pandemic in Italy: updated estimates indicate persistent excess in recent months. Medicina Lavoro 2022, 113, e2022021. https://doi.org/10.23749/mdl.v113i2.13108.
Author Response
Thank you for revewing our manuscript and for the valuable comments and suggestions. Please find enclosed a point-by-point response to the comments.

Round 2
Reviewer 3 Report
Thank you to the authors for strengthening their submission in the manner suggested by this reviewer. Each of the changes that have been made have improved the submission in the manner expected.
There are now only minor corrections to be made to the paper with respect to the following: citing an additional reference in the introduction, one change of wording, and a number of punctuation changes. These punctuation changes weren’t mentioned by this reviewer in the first review; however, at this point, they need to be corrected.
There are a number of instances where a hyphen has been used in place of an em dash or an en dash. Hyphens are used to join words and cannot be used to replace em dashes (used in place of semi-colons) or en dashes (used to join numbers). Please change all instances of where a hyphen is used inappropriately to the em dash or the en dash (depending on the situation). Please note, there are no spaces before or after an em or en dash.
Line by line suggested edits
21 Change “of Italian” to “of the Italian”.
24 Change “2015-2019” to “2015–2019”.
47 Thank you to the authors for adding the information regarding COVID-19. Please provide a reference for this information.
88 Change “2011-2021” to “2011–2021”.
69 Change “, 35,000” to “; 35,000”.
102 Change “2015-2019” to “2015–2019”.
104 Change “2015-2019”to “2015–2019”.
109 Change “2011 to 2019 (plus January-February” to “2011–2019 (plus January–February”.
125 Table 1 is mentioned here. Table 1 should follow this paragraph in which it is first mentioned, after line 138 with Figure 1 then following Table 1.
146 Change “of -respectively - all-cause” to “of—respectively—all-cause”.
148 Change “253 - 546” to “253–546”.
149 Change “- thus” to “—thus”.
150 Change “5.0-10.5” to “5.0–10.5”.
151 Change “65-79 years, and 5.9% (95% CI: 3.6-8.3)” to “65–79 years, and 5.9% (95% CI: 3.6–8.3)”.
152 Change “(95% CI: 3.8 – 10.6) among males and 5.4% (95% CI: 2.2 – 8.5)” to “(95% CI: 3.8–10.6) among males and 5.4% (95% CI: 2.2–8.5)”.
156 Change “- 19.6” to “–19.6”.
158 Change “(-6.1%; 95% CI: -9.9– -2.3)” to “(-6.1%; 95% CI: -9.9– -2.3)”.
159 Change “-14.0 – -4.0), than in males (-1.9%; 95% CI: -7.9 – 4.0” to “-14.0–-4.0) than in males (-1.9%; 95% CI: -7.9–4.0”.
161 Change “5.5 – 23.3) and females (26.3%; CI: 10.5 – 36.8” to “5.5–23.3) and females (26.3%; CI: 10.5–36.8”.
162 Change “-15.9 – -5.8” to “-15.9– -5.8”.
168 Change “2015-2019” to “2015–2019”.
Table 1
Change “2015-2019” to “2015–2019”
In both the “Excess deaths (95% C.I.)” and the “% excess deaths (95% C.I.)” column, please ensure that each of the en dashes used have no space before or after the en dash.
177-178 Figure 1
Please increase the size of Figure 1 to fit the entire width of the page (similar to the Tables). The font of Figure 1 is too small to read.
184 Change “2011-2019” to “2011–2019”.
Table 2
In both the “Excess deaths (95% C.I.)” and the “% excess deaths (95% C.I.)” column, please ensure that each of the en dashes used have no space before or after the en dash.
Table 3
In both the “Excess deaths (95% C.I.)” and the “% excess deaths (95% C.I.)” column, please ensure that each of the en dashes used have no space before or after the en dash.
193 Change “2011-2019” to “2011–2019”.
202 Please use square brackets for the citations, not round brackets.
224-225 Change “2015-2019” to “2015–2019”.
226 Change “2015-2019” to “2015–2019”.
232 Please put the first “the” back that has been deleted from the sentence”
236 Change “2015-2019” to “2015–2019”.
242 -244 Change “Italy – least affected by the pandemic in 2020 – the 2021 excess mortality was over 13%, in the North – hit first by the pandemic in 2020 [2,20,28] – 2021” to “Italy—least affected by the pandemic in 2020—the 2021 excess mortality was over 13%, in the North—hit first by the pandemic in 2020 [2,20,28]—2021”.
Author Response
Thank you so much for every suggested edits, we corrected all of them. In addition, we changed the Figure. Please let us know if it is acceptable as in the current form.